# Boron/Difluoroamino (B/NF_2_) Composites Prepared Through an Energetic Fluorinated-Centerd Surface Modification Strategy to Enhance Their Ignition and Combustion Characteristics

**DOI:** 10.3390/nano14221772

**Published:** 2024-11-05

**Authors:** Junqi He, Jing Lv, Yanan Li, Wenfang Zheng, Renming Pan

**Affiliations:** 1School of Safety Science and Engineering (School of Emergency Management), Nanjing University of Science and Technology, Nanjing 210094, China; hejunqi66@sina.com (J.H.); liyanankyzy@yeah.net (Y.L.); panrenming@njust.edu.cn (R.P.); 2School of Chemistry and Chemical Engineering, Nanjing University of Science and Technology, Nanjing 210094, China; lvjing9487@163.com

**Keywords:** NF_2_, B/NF_2_ composites, surface modification, coating, ignition and combustion

## Abstract

To enhance the ignition and combustion characteristics of boron (B), in this study, a suitable, energetic fluorinated group (NF_2_) that can improve energy and promote combustion efficiency was utilized and B/NF_2_ composites (B/PDB) with three different particle sizes (10–20 μm, <5 μm, and 0.5–2 μm) were prepared through energetic fluorinated surface modifications with a PDB layer, a copolymer of difluoroaminomethyl-3-methylethoxybutane and 3,3′-bis(azidomethyl)oxetane, coated on the surface of B. The morphology and structure of B/PDB were characterized via the FTIR, SEM, TEM, and XPS techniques. The results indicate that all B/PDB particle sizes were successfully coated by NF_2_ on the surfaces of B particles through the PDB layer. The TG curves in the thermal analyses were used to determine the amount of the PDB layer of B/PDB with different particle sizes. Based on the DSC curves, NF_2_ of composites with better catalysis during ammonium perchlorate (AP) decomposition. Additionally, the effects of NF_2_ on both B/PDB and B/PDB with AP were investigated through PY-GC/MS, ignition, and combustion. Compared with pure B, NF_2_ significantly improved the thermal conductivity, thereby decreasing the ignition delay of B/PDB, and the ignition delay of B/PDB with AP. The combustion of B/PDB and AP was more intense, extending the combustion duration, forming volatile fluorine compounds, and increasing combustion reaction efficiency. In general, this energetic fluorinated-centred surface modification has potential applications to enhance the ignition and combustion characteristics in B.

## 1. Introduction

Good release of solid propellant energy usually depends on the ignition and combustion characteristics of the fuel in practice [1,2,3]. Boron (B) is a promising fuel because it can theoretically provide high gravimetric heat (58.30 MJ/kg) and volumetric heat (136.44 kJ/cm^3^) [4,5] during combustion. The thermal transfer between B and the outside environments is hindered by its high melting (2075 °C) and boiling points (4000 °C), along with the oxide layer on the surface, which has a low melting (~450 °C) and high boiling points (~2042 °C) [6,7,8]. Consequently, the low thermal conductivity of B leads to ignition difficulties and low combustion efficiency and hinders energy release during the combustion of B and other components in the solid propellant [9,10].

Some functional materials are commonly coated onto the surface of B as a surface modification method to synthesize boron composites and thus enhance the performance of B with no drastic change in morphology [11,12,13]. Depending on its specific materials, a coating can enhance the specific surface area of fuel particles and improve surface reactivity. Energetic compounds such as ammonium perchlorate (AP) [14], KNO_3_ [15], and nitroguanidine (NQ) [16] have been used to coat B. These compounds can increase thermal energy to raise the combustion temperature quickly for B, thus enabling B and B_2_O_3_ to volatilize at high temperatures. Internal B can then continue to react to improve combustion efficiency [17,18]. Fluorinated (F) materials such as polytetrafluoroethylene (PTFE) [19,20], polyvinylidene fluoride (PVDF) [21,22,23,24] and polytrifluorochloroethylene (PCTFE) [25], have a better role as coatings on B surface than energetic materials, themselves and decomposition products can react with B to mainly form BF_3_ with a boiling point of only −100.15 °C, thus eliminating the oxide layer with lower temperature and making B more volatile during combustion [26]. This process increases the speed of thermal transfer for B. In addition, the energy released by the combustion of fluoride with boron particles to produce BF_3_ is no less than that released by the reaction to produce B_2_O_3_. Hence, fluorinated materials generally improve the combustion efficiency of B better than energetic compounds. For the above reasons, fluorinated materials used for B coatings have attracted significant attention [27,28]. Regrettably, using these fluorinated materials coatings without any energetic group may result in a loss of B’s high gravimetric heat [2,29,30,31,32]. However, energy and combustion efficiency are equally important for a solid propellant. Building on the above statement, it is desirable to coat B particles with energetic fluorinated groups as this process combines all functions of the above two methodologies. Nevertheless, research on more concise and efficient methods for enabling target groups to fulfil the above requirements on the surfaces of B particles remains incomplete.

Difluoroamino (NF_2_) [33] has a theoretical density of 2.303 g/cm^3^ and an enthalpy of formation of −32.7 kJ/mol. The N–F bond has higher reactivity than the C–F bond, with the F in NF_2_ appearing in the form of the oxidant. Related compounds introducing synthesized NF_2_ can significantly enhance energy levels and thus be used as potential oxidants or binders in solid propellants [33,34]. Notably, the HF produced by NF_2_ during combustion has a low relative molecular mass, with the bond energy of HF reaching up to 565 kJ/mol [35,36]. Meanwhile, the F in NF_2_ serves as a good combustion aid and is as effective as the aforementioned fluorinated polymers in improving the combustion efficiency of B particles. Consequently, NF_2_ combines the functions and advantages of both F elemental and energetic groups. Thus, the emergence of NF_2_ offers a novel solution to the aforementioned issues if it is incorporated on the surfaces of B particles to synthesize B/NF_2_ composites. PDB, a copolymer of difluoroaminomethyl-3-methylethoxybutane and 3,3’-bis(azidomethyl)oxetane, is a type of difluoroamino (NF_2_) polymer with additional azide groups and can facilitate interactions with the surfaces of B particles [37]. Thus, PDB serves as a carrier for NF_2_ and is coated on the surface of B using a surface method that does not destroy these functional groups.

B particles were handled using surface pretreatment to enable the surfaces of B particles to come into contact with PDB. Due to its strong reactivity, NF_2_ can quickly facilitate a reaction. In this study, the surface of B is linked to the telohydroxyl group of PDB through surfactants and soft chemical methods. Isocyanate curing agents are ultimately provided with PDB bonded on the surface of B to coat B with NF_2_ using the PDB layer. We successfully incorporated NF_2_ onto the surface of the B particle with three different particle sizes (10–20 μm, <5 μm, and 0.5–2 μm), leading to the synthesis of a B/NF_2_ composite, B/PDB. The structure and morphology of the B/PDB composites were also investigated. We found that the surface modification of NF_2_ significantly improved the overall performance of B particles. Importantly, this result verifies the potential of using NF_2_ for the ignition of B/PDB, as well as the catalytic behaviour of B/PDB on AP.

## 2. Experiment and Characterization

### 2.1. Materials

Boron (B) particles with diameters of approximately 10–20 μm (99%, amorphous), <5 μm (98%, amorphous), and 0.5–2 μm (99.9%, amorphous) were supplied by Macklin (Shanghai, China). Ethanol (95%), ethyl acetate, and hexyl hydride, all of analytical purity, were purchased from Wohua Chemical Co., Ltd. (Shanghai, China). In addition, (3-aminopropyl) and triethoxysilane (KH-550), both analytical grades, were purchased from Sinopharm Chemical Reagent Co., Ltd. (Shanghai, China). Analytically pure toluene diisocyanate (TDI) was purchased from Aladdin (Shanghai, China). PDB was synthesized and purified using Li’s method [37]. The molecular mass (*M_n_*) of PDB was ~3200 g/mol, with a polydispersity index (*M_w_*/*M_n_*) of 1.55. Ammonium perchlorate (AP) was provided by the Xi’an Modern Chemistry Research Institute, with a particle size of approximately 150 μm.

### 2.2. Pretreatment of B Powder

Pretreatment of the B particles involved adding a B particle of each particle size (10–20 μm, <5 μm, and 0.5–2 μm) to ethanol with reflux at 80 °C for 4 h for preliminary surface preparation. These particles were subsequently filtered and dried in a vacuum oven at 60 °C for 12 h. The above-treated B particles were thoroughly dispersed into ethanol, and KH-550 was added. After undergoing ultrasonic dispersion for 1 h, the B particle suspension was magnetically stirred at 60 °C for 2 h under a nitrogen atmosphere (N_2_). Following the reaction, each B particle was filtered and dried in a vacuum oven at 60 °C for 12 h.

### 2.3. Preparation of B/PDB by Solvent/Non-Solvent Method

In this study, B/PDB was prepared at various particle sizes using the solvent/non-solvent method. Figure 1 illustrates a diagram of the preparation process. In the typical procedure, 1 g of a pretreated B particle was dispersed in 50 mL of ethyl acetate, and PDB with 20% wt was added. This process was followed by ultrasonic dispersion for 30 min. The above mixture was then magnetically stirred at 60 °C for 2 h under N_2_. Next, the mixture was slowly added to 250 mL of hexyl hydride with TDI using vigorous, magnetic stirring. Notably, the temperature of hexyl hydride was 60 °C, and the addition speed of the mixture was 1 mL/min. Following the addition of the mixture, the temperature was reduced to room temperature (20 °C). The precipitated PDB adhered to the surface of each B particle and formed the desired coating layer with the aid of TDI. After adding the mixture, the reaction proceeded for 2 h at room temperature. Subsequently, the coated B particle was filtered, washed twice with hexyl hydride, and dried at 60 °C under a vacuum. Notably, the coating technology was similar for each particle size.

### 2.4. Characterisation and Methods

The Fourier transform infrared (FTIR) absorption spectra were measured using an NICOLETIS 20 spectrometer (Thermo Fisher Scientific, Waltham, MA, USA) in the range of 4000 to 500 cm^−1^. The micrographs were observed using an FEI QUANTA 400 FEG field emission scanning electron microscope (FESEM) (Oxford Instruments, Abingdon, Oxfordshire, UK) and a JEM-2100F conventional transmission electron microscope (TEM) (JEOL, Akishima, Tokyo, Japan). The X-ray photoelectron spectroscopy (XPS) measurements were conducted on an ESCALAB 250Xi spectrometer (Thermo Fisher Scientific, Waltham, MA, USA).

The thermal weightlessness of B and B/PDB was examined using thermogravimetric analysis (TGA/SDTA851E, Mettler Toledo, Zurich, Switzerland). The test was conducted under a N_2_ atmosphere at a heating rate of 10 °C/min within a temperature range of 50–500 °C. The weight of each sample ranged from 0.6 to 0.8 mg. The exothermic characteristics of the mixed samples of B and AP, B/PDB, and AP were studied via differential scanning calorimetry (DSC) using DSC3 (Mettler Toledo, Zurich, Switzerland). The test was conducted in an N_2_ atmosphere with a heating rate of 10 °C/min at a temperature range of 50–500 °C. In addition, the mass ratio of AP to B (or B/PDB) was 9:1, and the sample mass was approximately 0.6–0.8 mg. The volatile compounds of AP and B/PDB were analyzed using pyrolysis-gas chromatography/mass spectrometry (PY-GC/MS) at 300 °C and 500 °C, respectively. The samples were mixed with a mass ratio of 9:1 for AP:B/PDB. For pyrolysis, approximately 0.2 mg of mixed samples was placed in a Frontier 3030D pyrolizer interfaced with a Shimadzu QP2010ULTTA GC/MS (Kyoto, Japan). Helium was used as the carrier gas for this analysis. The pyrolysis and GC/MS interfaces were kept at 300 °C, and the GC was temperature-programmed from 40 °C (for 2 min) to 140 °C at 10 °C/min and from 140 °C to 300 °C at 20 °C/min. The MS was operated in the electron impact mode (70 eV) for *m*/*z* 10–550.

Approximately 20 mg of the B or B/PDB samples for ignition was weighed and placed into a quartz tube (diameter: 3 mm, length: 5 mm). The mixed samples of AP and B or B/PDB had a mass ratio of 8:2. Around 20 mg of each mixed sample was weighed and placed into a quartz tube with the exact aforementioned specifications. After combustion of the mixed samples, the combustion residues were collected. Then, the micrographs were characterized via FESEM, as mentioned above. The samples were ignited using a laser. The laser ignition device was a 500 W fibre laser produced by Nanjing University of Science and Technology with a supply voltage of 220 V ± 10%. The laser power used in this experiment was 40 W. In addition, the laser spot had a diameter of 5 mm, and the duration of light emission was 1 s. Images of the ignition and combustion processes were recorded with a high-speed camera (Photron, FASTTXAM Mini UX100, Tokyo, Japan) at 1000 fps.

## 3. Result and Discussions

### 3.1. Characterization of Structure and Morphology

Figure 1 depicts the FTIR spectra of B/PDB with 10–20 μm, <5 μm, and 0.5–2 μm. The spectra were compared for B without a coating and PDB. The FTIR spectra indicate significant changes in the B/PDB with different particle sizes compared to the results for pure B particles. Overall, the characteristic peaks of all B/PDB were identified. The symmetric telescopic vibration of N_3_ was observed at nearly 2095 cm^−1^ (Figure 1a,c,e) [37,38], and the N_3_ symmetrical telescoping vibration was observed at nearly 1297 cm^−1^ (Figure 1b,d,f) [37,38], demonstrating that N_3_ still existed in B/PDB of all sizes after the coating reaction. The characteristic peak at nearly 1100 cm^−1^ is due to the C-O-C in the main chain segment of the binder [36,37]. The weak characteristic peaks at nearly 1015 cm^−1^ and 900 cm^−1^ (Figure 1b,d,f) are attributed to N–F bond vibrations [35,36,37]. Meanwhile, the characteristic peak at nearly 800 cm^−1^ (Figure 1b,d,f) is attributed to C–N bond vibrations [35,36,37]. The characteristic peaks of NF_2_ are not very pronounced and remain affected by B particles. However, these peaks provide evidence for the presence of NF_2_ on the surface of B. Nevertheless, this result requires further characterization.

The morphology of B and B/PDB particles with different particle sizes was also analyzed with FESEM. Figure 2 illustrates these morphologies. As depicted in Figure 2d–f, all B/PDB samples exhibit disordered atoms, with PDB cladding stacked on the surface. The B/PDB surfaces are smoother and cleaner than the rougher B surfaces, which have sharp protrusions. A typical EDS mapping analysis was performed to further investigate the distribution of PDB on the B particles. The EDS element mapping results (Figure 2g–l) for B/PDB and B reveal the generation of N and F elements after coating; these elements were found to be evenly distributed on the B sphere. The combined results of the FTIR spectra demonstrate that NF_2_ retained its original structure and became uniformly distributed on the surfaces of B particles through successful coating with the PDB carrier.

Figure 3 presents a TEM image of B/PDB, and Appendix A shows the corresponding EDS images. As indicated in Figure 3(b-1,b-2,d-1,d-2,f-1,f-2), B/PDB presents a typical core–shell structure, with the transparent film-like substance PDB adhering to the periphery of the B particles. Based on the morphological characterization, NF_2_ is distributed on the B/PDB surface, and the thin coating layers of PDB form core–shell structures. We also validated and complemented the FTIR spectra of the B/PDB using TEM images.

Figure 4 presents the XPS spectra of B and B/PDB particles with different particle sizes. These spectra were measured to analyze the elements and states of the surface. The B powder primarily consists of B, O, and C, in which the O and C elements result from the oxide layer. More evident peaks of N and F elements can be observed at 10–20 μm in B/PDB, <5 μm in B/PDB, and 0.5–2 μm un B/PDB, which suggests the presence of the PDB coating layer on the surfaces of the B particles. The spectrum of C 1s (see Figure 4c–e) for B/PDB presents signal peaks at the ~284.4 eV, 284.8 eV, ~286.2 eV, ~286.6 eV, and ~289 eV positions after splitting, corresponding, respectively, to C–H [39], C–C [39,40,41], C–O [39,40,42], C–N [41], and C=O [39,40,41,43] bonds. Meanwhile, the O 1s spectra (Figure 4f–h, respectively) of B/PDB presents signal peaks at 532.5–533.3 eV, 531.5–532 eV, and ~533 eV, which correspond to O–B [17,27,44], O–C [39,44], and O=C [39,44] bonds, respectively. Combined with an analysis of the N element (Figure 4i–k) with two consecutive peaks, the first peaks at 399.4 eV and 400.4 eV belong to –CO–NH– [42,45] and N_3_ [42], respectively. The existence of –CO–NH– confirms the occurrence of a curing reaction for PDB with the isocyanate groups of toluene diisocyanate on the surfaces of B particles during the coating process. The other peak at ~404.15 eV indicates the establishment of N–F. As depicted in the N 1s (~404.15 eV) [43] and F 1s (~686.55 eV) [46] spectra, elemental F still exists in the form of NF_2_, and the structure of the N–F bond was not damaged. Combining the results of the FTIR, FESEM, and TEM analyses shows that N_3_ and NF_2_ were not destroyed and became routinely encapsulated through PDB on the surfaces of B particles through curing with toluene diisocyanate.

### 3.2. Characteristics of Surface Coating

As depicted in Figure 5, the actual amount of PDB coating can be roughly measured through PDB thermal decomposition via TG analysis. The decomposition of PDB started at nearly 200 °C mainly due to N_3_ and NF_2_ [35,36,37], leading to a mass loss of approximately 11.58 wt% (10–20 μm), 9.15 wt% (<5 μm), and 12.16 wt% (0.5–2 μm) in B/PDB. The actual coating of NF_2_ is lowest with a particle size of <5 μm (less than 10%), while the other two particle sizes are close to each other. Here, the amount of the actual NF_2_ coating of B/PDB (0.5–2 μm) is slightly higher than composites with a particle size of 10–20 μm, despite having a smaller particle size. This phenomenon is attributed to the surface properties and surface energy of B particles with different particle sizes.

Figure 6 illustrates the ignition and combustion processes of B and B/PDB particles with different particle sizes. Appendix A presents the statistics of the ignition delay, while Appendix A presents the combustion duration. Figure 6 shows that the ignition delay was reduced by NF_2_ after PDB became attached to the surfaces of the B particles. In Figure 6, the ignition delay decreased by 126 ms (10–20 μm), 141 ms (<5 μm), and 133 ms (0.5–2 μm). Notably, NF_2_ increased the flame intensity, and all B/PDB samples with different particle sizes emitted the green flame characteristic of B. Moreover, the combustion duration of B/PDB was shorter than that of B without an NF_2_ coating at the same particle size. Figure 6 shows that the intense combustion starts at 57 ms (10–20 μm), 46 ms (<5 μm), and 28 ms (0.5–2 μm) and is accompanied by the creation of an obvious green flame produced after ignition. These findings suggest that NF_2_ can assist in the combustion processes of B particles, enable the rapid thermal transfer of B/PDB particles, and accelerate the combustion reaction of B. By enabling B/PDB to quickly combust, the combustion duration can be reduced. In this way, coating the B particle surface with NF_2_ promotes the combustion efficiency of B and enhances the corresponding ignition and combustion characteristics. With a decrease in particle size, the ignition delays of both B and B/PDB decreased together, and the combustion intensity improved. Furthermore, after coating with NF_2_, 0.5–2 μm B/PDB was found to optimally facilitate the minimum ignition delay.

### 3.3. NF_2_ Effect of Catalytic Behaviour on AP and B Particles

Ammonium perchlorate (AP) is a vital oxidant component in solid propellants. The burning rate of a propellant is closely linked to the decomposition of AP [39]. Numerous studies have demonstrated that a reduction in the decomposition temperature of AP can increase the burning rate of the propellant. The catalytic effects of B and B/PDB samples of various powder sizes on the decomposition of AP were analyzed via DSC. Figure 7 presents the corresponding DSC curves. As indicated in Figure 7, the thermal decomposition of pure AP consists of the following three stages: an endothermic peak at 243.3 °C, which is attributed to the crystal transition, and two exothermic peaks at 298.5 °C and 403.1 °C, which correspond, respectively, to the low-temperature and the high-temperature decomposition (THTD) of the AP exothermic process. After adding B-containing samples without NF_2_ coating for AP, the intensity of all peaks, both endothermic and exothermic, decreased significantly. Additionally, the THTD of AP increased significantly. The peaks of THTD occurred at 425.5 °C (10–20 μm), 415.0 °C (<5 μm), and 415.8 °C (0.5–2 μm), respectively, in the absence of a catalytic effect from NF_2_. The aforementioned results suggest that B plays a limiting role in the decomposition of AP. In contrast, after adding B/PDB containing NF_2_ for decomposition, the THTD peaks of AP occurred at 384.5 °C (10–20 μm), 373.9 °C (<5 μm), and 368.2 °C (0.5–2 μm). Compared to the results without NF_2_, THTD was reduced by 41 °C (10–20 μm), 41.1 °C (<5 μm), and 47.6 °C (0.5–2 μm), respectively. Meanwhile, with a decrease in THTD, the high-temperature exothermic peaks of AP catalyzed via B/PDB shift closer to the low-temperature exothermic peaks and become a continuous set of exothermic peaks. This result suggests that the exothermic nature of AP changed from two separate phases to two consecutive phases or nearly one phase. A more concentrated exothermic reaction favors further energy release between AP and B in a solid propellant. The B/PDB additives can decrease the decomposition temperature of AP and promote the thermal decomposition of AP at high temperatures. Moreover, the catalytic effect for AP increases with a decrease in particle size.

Based on the aforementioned DSC results and discussions, we analyzed the main pyrolysis volatiles of the mixed samples consisting of AP and B/PDB (10–20 μm, <5 μm, and 0.5–2 μm) using a PY-GC/MS system at 300 °C and 500 °C, respectively. Chromatograms for the mixed-sample pyrolysis are presented in Appendix A. The majority of peaks were identified using the NIST library and previous studies [47,48]. The main compounds identified are listed in Appendix A. As the pyrolysis was performed at 300 °C, the pyrolysis products of B/PDB and AP were dominated by compounds composed of the elements C, H, N, and O such as ethenyl formate (C_3_H_4_O_2_) 2-vinyl-4,5-dihydrooxazole (C_5_H_7_NO) and vinyl benzoate (C_9_H_8_O_2_). The chlorine compounds produced were predominantly chlorine (Cl_2_) and hydrogen chloride (HCl). With a decrease in the particle size, a small amount of silicone compounds was found to exist in the pyrolysis products of AP and B/PDB with <5 μm and 0.5–2 μm particle sizes. This result is likely due to the strong reactivity of B/PDB at these particle sizes, which corroded the tubes of the pyrolizer during the pyrolysis reaction with AP, resulting in, e.g., phenethyl(triethylsilyl) ether (C_14_H_24_OSi), *N*-methyl nicotinimidate, *O*-trimethylsilyl (C_10_H_16_N_2_OSi). Notably, B/PDB with a 0.5–2 μm particle size formed the fluorinated compound benzoic acid, 2,2-bis(trifluoromethyl)-1-aziridinyl ester (C_11_H_7_F_6_NO_2_) during pyrolysis with AP due to its stronger reactivity at 300 °C. When pyrolysis occurred at 500 °C, the main products contained C, H, N, and O, including benzoic acid (C_7_H_6_O_2_), methyl nitrate (CH_3_NO_3_), and 3-nitro-1-phenylpropan-1-one (C_9_H_9_NO_3_). In parallel, chlorine compounds such as HCl, Chlorobenzene (C_6_H_5_Cl), and 1,2-Dichlorobenzene (C_6_H_4_Cl_2_) were observed during pyrolysis at 500 °C. As the pyrolysis temperature increased, some fluorine compounds and silicone compounds of AP and B/PDB were also detected. The pyrolysis of 10–20 μm B/PDB with AP yielded trimethylfluorosilane (C_3_H_9_FSi), 2-(3,3,3-trifluoropropyl)piperidine (C_8_H_14_F_3_N), benzoic acid,2,2-bis(trifluoromethyl)-1-aziridinyl ester (C_11_H_7_F_6_NO_2_), 4′-methoxy-3-(3,4,5-trifluorophenyl)propiophenone (C_16_H_13_F_3_O_2_), and silicone compounds such as Hexamethylcyclotrisiloxane (C_6_H_18_O_3_Si_3_). Compounds such as fluoro(trinitro)methane (CFN_3_O_6_), 2,4-difluorobenzonitrile (C_7_H_3_F_2_N), and 2-(3,3,3-trifluoropropyl)piperidine (C_8_H_14_F_3_N) were generated in the pyrolysis products of AP and <5 μm B/PDB. Notably, B/PDB with a particle size of 0.5–2 μm generated the fluorine compounds and silicone compounds listed in Appendix A with AP under pyrolysis at 500 °C. Here, the organic volatiles generated via AP and B/PDB pyrolysis are dominated by compounds composed of the elements C, H, N, and O at 300 °C and 500 °C, respectively. Compared with the pyrolysis products at 300 °C, fluorine compounds were more likely to be generated at 500 °C. In addition, all particle sizes of B/PDB contained fluorine compounds after pyrolysis with AP. In addition, the higher reactivity corroded the tubes of the pyrolizer, which also led to the generation of silicone compounds.

Based on the results above and data for DSC and PY/GC-MS, further ignition and combustion experiments were conducted on mixed samples of AP and B to investigate the energy released from the combustion of the common oxidant AP. The ignition tests and combustion processes for these mixed samples were recorded using a high-speed camera, with the pictures presented in Figure 8. The corresponding ignition delay and combustion duration statistics are shown in Appendix A, respectively. Overall, the ignition delays of all mixed samples of AP and B (or B/PDB) were short; the maximum was only 22 ms, unlike the case with the laser ignition of B alone. The ignition delays of the mixed samples of AP and B/PDB coated by NF_2_ were shorter than those of the mixed samples without NF_2_, although the decrease was not significant. In contrast, the combustion durations for mixed samples of AP and B/PDB coated with NF_2_ increased dramatically. Moreover, a distinctive boron-specific green flame was observed during combustion, which was not observed among the samples without an NF_2_ coating. Notably, the mixed sample of 10–20 μm B had a short combustion duration. However, the mixed sample of B/PDB with the same particle size increased the combustion duration from 453 ms to 983 ms (nearly double) and decreased the ignition delay from 22 ms to 11 ms. At the same time, the intensity of the trailing flame during the combustion process was considerably improved. Furthermore, the NF_2_ on the surfaces of B particles enhanced the energy released during the combustion process with AP. In addition, the energy release of the mixed sample with <5 μm B/PDB was improved via NF_2_ during the combustion process using AP, with the ignition delay decreased by 6 ms, and the combustion duration extended by 591 ms. Additionally, after coating with NF_2_, the ignition delay for the mixed sample of B/PDB (0.5–2 μm) decreased from 18 ms to 5 ms, and the combustion duration increased from 651 ms to 1072 ms. The NF_2_ of B/NF_2_ composites in the mixed samples similarly increased B’s internal thermal conductivity, allowing the mixed samples to transfer heat quickly and successfully ignite, decreasing the ignition delay and increasing the overall combustion efficiency.

The combustion residues of B with AP adhering to the quartz tube wall were also investigated to further understand the reaction process and changes after ignition. These burned-out particles were collected and analyzed via SEM and EDS. The SEM graph is shown in Figure 9, and the results of EDS are illustrated in Appendix A. According to the results of previous studies [49,50], the oxide layers on the surfaces of B particles assume a viscous liquid state during the combustion process due to their high boiling points. At the end of combustion, these particles become tightly aggregated due to binding of the viscous layer on the surface. This process results in a significantly clustered combustion product of B and AP, resulting in low combustion efficiency. As shown in Figure 9, for B without NF_2_ on its surface, the combustion of B with AP after ignition produced a large amount of agglomerates, as described above. These agglomerates contained large quantities of incompletely combusted B. The above results indicate the oxidation of incomplete B particles in the combustion reaction. Combined with the results for the characterization of EDS, the main components of the above residues were B_2_O_3_, incompletely combusted B particles, and C-containing particles. When NF_2_ present on the surface of B with different particle sizes was inserted into the combustion reaction with AP after ignition, the produced residues showed significant changes compared to those produced using B without a coating. Although the residues still agglomerated, the particle sizes were still significantly reduced compared to those when no ignition occurred. This result suggests that the NF_2_ surface on B improved the combustion and reaction efficiencies with AP, which corresponds to the combustion ignition diagrams in Figure 8. In this case, the combustion of these residues with particle sizes of 10–20 μm and 0.5–2 μm was greater than <5 μm, which also correlates with the actual amount of PDB coating characterized via TG (Figure 5). This result is mainly due to the high reactivity of NF_2_ on the surface of B, which reacts violently with AP and B on the surface along with N_3_, increasing the combustion reaction energy. In this way, the B combustion reaction process enabled high temperatures to be reached more quickly due to the volatilization of B_2_O_3_. At the same time, the generated BF_3_ gas removed more B_2_O_3_ droplets on the surface, enabling the internal B to continue the combustion reaction and form a large number of surface voids and small particles.

## 4. Conclusions

In summary, an energetic fluorinated-centred surface modification with NF_2_ coated on the surfaces of B particles was successfully achieved using the PDB layer, leading to B/NF_2_ composites (B/PDB) with different particle sizes being synthesized. The corresponding characterization demonstrated that the NF_2_ of composites with no damage and a PDB layer on the surface of B were 11.58 wt% (10–20 μm), 9.15 wt% (<5 μm), and 12.16 wt% (0.5–2 μm). Moreover, B/PDB presented better catalytic behaviour than B on AP based on the DSC curves. Compared to pure B, the high-temperature decomposition of AP’s exothermic process decreased by more than 40 °C with B/PDB. We also analyzed the improvements to B/PDB’s thermal conductivity, ignition, and combustion characteristics. Further, the ignition delay of B/PDB decreased, with the most presenting a delay of 48 ms (10–20 μm), and the green flame produced by B was more obvious than that of pure B. The inclusion of NF_2_ in the ignition and combustion of B/PDB and AP improved the combustion efficiency of B based on the presence of PY-GC/MS and combustion residues, indicated by significantly reduced fluorinated gas compounds and residue particle sizes. More impressively, the ignition delay of B/PDB and AP decreased, with the most presenting a delay of 13 ms (0.5–2 μm). Furthermore, the combustion duration was extended, with the most durations being 768 ms (<5 μm). Ultimately, this study described an energetic fluorinated surface modification strategy with synthesized B/NF_2_ composites and highlighted the positive impacts of these composites.

## Data Availability

The data presented in this study can be requested from the corresponding author.

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
