# Peer review of "Boron/Difluoroamino (B/NF2) Composites Prepared Through an Energetic Fluorinated-Centerd Surface Modification Strategy to Enhance Their Ignition and Combustion Characteristics"

_nanomaterials, 2024, doi:10.3390/nano14221772_

Round 1
Reviewer 1 Report
Comments and Suggestions for Authors
This manuscript explores a novel surface modification strategy using NF2-based fluorinated coatings on boron particles to enhance ignition and combustion efficiency. Given the challenges of achieving efficient combustion with boron due to oxide-layer formation, the attempt to combine energetic and fluorinated functionalities is commendable. Moreover, the investigation into particle size effects on ignition delay and combustion performance adds depth to the work. The study makes a meaningful contribution by addressing key obstacles in solid propellant design and showing how the NF2 layer can reduce ignition delays and boost combustion, especially when combined with ammonium perchlorate (AP). With additional discussion around the novelty and a few methodological improvements, this work has the potential to make a strong contribution to the field of high-performance solid propellants. I recommend considering the manuscript for publication at Nanomaterials after addressing the following minor revisions:
1. While the manuscript offers new insights, the reported approach draws on concepts explored in prior studies using fluoropolymers such as PVDF and PTFE coatings to enhance boron combustion (e.g., 10.3390/molecules28073209, 10.1007/s10853-024-10203-8). The authors should elaborate on what makes NF2 superior to these established alternatives. Other fluorinated coatings are known to improve oxidation resistance and combustion efficiency, sometimes in combination with catalysts like CuO or metal oxides (e.g., 10.1134/S0010508224010131). A clearer comparison would solidify the novelty claim of the NF2-based modification.
2. The manuscript discusses the impact of NF2 on ignition and combustion but leaves room for deeper mechanistic insights. Specifically:
- How does the decomposition behavior of NF2 compare with other fluorinated materials at high temperatures?
- Does NF2 generate specific combustion products (e.g., BF3 or HF) that directly influence the reactivity of boron during combustion?
Including additional pyrolysis or kinetic data would clarify these points and align with the study’s objectives.
3. Although the manuscript compares bare boron with NF2-coated composites, adding a more comprehensive control (e.g., boron with non-energetic fluorinated coatings) would better demonstrate the advantages of NF2. The scalability and reproducibility of the coating process are also essential to consider—what challenges might arise in scaling this technique, and how does NF2's performance change under different environmental conditions (e.g., high humidity)?
4. While the experimental results are promising, including error bars and statistical analysis for key measurements such as ignition delay and combustion duration would improve the reliability of the findings. For example, multiple trials for thermal decomposition and ignition experiments with average values and standard deviations would strengthen the conclusions.
Author Response
- In the introduction, it is explained that the main reason why NF2is superior to other fluorinated polymers is that it has F elements and also has energetic properties, and the HF produced during pyrolysis can also release higher energy in addition to providing F elements to help the B combustion efficiency at the same time. The C-F bonds such as PVDF and PETE are unable to realize the above at the same time.
- The thermal behaviour of NF2is specifically described in reference 37: “DFAMO/BAMO copolymer as a potential energetic binder: Thermal decomposition study.”, which describes the thermal decomposition mechanism of the PDB. Different from other fluorinated polymers, NF2mainly produces HF during thermal decomposition, which is explained in the introduction. The effect of PDB on B/PDB and AP was analyzed by PY/GC-MS and the combustion residue of AP and B/PDB were analyzed. It was observed that the residue of B/PDB composites with 5 μm was different from that produced by B/PDB at the other two particle sizes, which was mainly said to since the composite particles with 5 μm had the least amount of PDB coating (the characterization results of TG).
- Due to the lack of relevant energetic materials, it was not possible to prepare relevant composite particles for comparison under the same conditions, and by comparing with other energetic coatings of B in the literature, it was achieved that B could be successfully ignited in an air environment and produce the characteristic green flame of B (in the literature, the ignition is mainly in the environment of high-purity oxygen). The Coating process has not yet been carried out for the synthesis of a larger dose of the preparation, and it is expected that in the scaling up of the experiments, suitable equipment will be needed to ensure the yield. Experiments with NF2in special environments have not yet been publicly investigated in depth.
- The relevant ignition data is attached in the Supporting Information in the form of error bars (Figure S2).

Reviewer 2 Report
Comments and Suggestions for Authors
The manuscript "Boron/difluoroamino (B/NF2) composites, prepared through an energetic fluorinated-centred surface modification strategy…" is devoted to the modification of micron-sized boron particles with a fluorine polymer coating to improve the ignition and combustion characteristics of boron. The manuscript is not directly related to the topic of the special issue "Thermal conductivity of nanomaterials and their applications" of the journal "Nanomaterials". This issue is aimed at presenting the role and application of nanomaterials in improving thermal conductivity. The results of the work showed a significant improvement in the combustion and ignition characteristics of boron particles when using a coating based on a copolymer of difluoroaminomethyl-3-methylethoxybutane and 3,3'-bis(azidomethyl)oxetane containing an active NF2 group. The results are important for the development of boron-based solid propellant compositions, but do not contribute to improving thermal conductivity.
The text requires corrections and additions.
Lines 107-108 contain the text - "After undergoing ultrasonic dispersion for 1 h, the Al powder suspension was magnetically stirred at 60 ℃ for 2 h under a nitrogen atmosphere (N2). Following the reaction, the Al powder was filtered and dried in a vacuum oven at 60 ℃ for 12 h." - aluminum was not used in the work, but Ammonium perchlorate (AP) was used. The properties of the AP used, its trade mark, particle size, etc. are not specified in the text.
The decision to publish the manuscript should be made by the editor of the special issue, in my opinion this manuscript is not suitable for the issue topic.
Author Response
Thank you very much for taking the time to review this manuscript and for your questions and suggestions. We have revised the manuscript accordingly in the resubmission based on your suggestions and have responded to the questions raised, thank you again for taking the time to review the revised manuscript.
Errors related to lines 107-108 in the original article have been corrected (line 113), and the manufacturer's information and particle size comments for AP have been added (lines 106-107). Additional discussion on the thermal conductivity of B/PDB is provided in the preface section and in the performance characterization.

Reviewer 3 Report
Comments and Suggestions for Authors
In this work, Boron/difluoroamino B/NF2 compositions with particle size variation between 0.5 and 20 microns have been synthesized using a solvent/non-solvent chemical route. The obtained composites were characterized by the TGA, FTIR, SEM, XPS, and burning rate techniques. The manuscript presents a good experimental procedure with properly discussed results. From my point of view, the manuscript can be accepted in this journal after minor corrections.
Some particular comments:
1. On page 5, line 186 the authors claim; “As depicted in Figure 2d, 2e, and 2f, all B/PDB exhibit amorphous particles,…”; the authors must define the amorphous particle. Amorphous particle is described commonly as disordered atoms.
2. In Figure 2, If the objective is to compare the samples, the authors must provide the SEM images of (a) B (10~20 μm); (b) B (<5 μm); (c) B (0.5~2 μm); (d) B/PDB (10~20 μm); (e) 195 B/PDB (<5 μm); (f) B/PDB (0.5~2 μm) at the same amplification. Here, a), b), and c) are at different amplifications concerning d), e) and f).
3. On page 6 line 198 the authors claim; “Figure 3 depicts the TEM image of B/PDB. As indicated in Figure 3b, 3d and 3f, B/PDB 198 presents a typical core-shell structure,…”. To validate this claim, the authors must provide elemental-mapping images by EDS.
Author Response
Thank you very much for taking the time to review this manuscript and for your questions and suggestions. We have revised the manuscript accordingly in the resubmission based on your suggestions and have responded to the questions raised, thank you again for taking the time to review the revised manuscript.
- Page 5, line 186 has been amended as suggested (line 195).
- Figure 2 has been reworked as suggested
- Since the particle sizes of both B and B/PDB are in the micrometre scale, TEM cannot get a clear image of the particle boundaries by EDS mapping, and can only get an EDS mapping image similar to that of SEM. We can only provide regular EDS images in Supporting Information (Figure S1).

Round 2
Reviewer 2 Report
Comments and Suggestions for Authors
1. The corrected text, as before, has no direct relation to the main topic of the special issue "Thermal conductivity of nanomaterials and their applications" of the journal "Nanomaterials". The text of the Conclusion now includes the new phrase "The improvement of B/PDB thermal conductivity, and enhancement of the ignition and combustion characteristics of B/PDB were studied.” However, there is no data on thermal conductivity in the manuscript.
2. It is necessary to make some minor corrections to the text. In the corrected text, the phrase "The corresponding ignition delay and combustion duration are statisticed for SX and SX, respectively" appeared - what are “SX and SX”?
Author Response
Regarding the doubt that the manuscript does not match the main topic of the special issue, here is the following explanation: B is difficult to volatilize at high temperatures due to its low melting point and high boiling point, and it continues to absorb thermal during the reaction process and cannot transfer the thermal. At the same time, the oxide layer (B2O3) generated by the reaction of B has a high melting point and high boiling point, which further hinders the thermal transfer between B and the outside environments, in which case the low thermal conductivity of B leads to the difficulty of ignition and low combustion efficiency. The B/NF2 composites prepared by coating NF2 make the B surface more thermal absorbent and volatile, and the thermal no longer accumulates and transfers quickly, resulting in the intuitive result of shortening the ignition delay of B, shortening the ignition delay of the mixed samples of B and AP, and enhancing the combustion.
